# Small-scale integrated farming systems can abate continental-scale nutrient leakage

**Gidon Eshel** [ID] *

Bard College, Annandale-on-Hudson, New York, United States of America

* geshel@gmail.com

**Data Availability Statement:** All relevant data are within the paper, its Supporting Information files, or in a publicly available data file whose DOI is 10.13140/RG.2.2.28878.18248.

**Funding:** The author received no specific funding for this work.

## Abstract

Beef is the most resource intensive of all commonly used food items. Disproportionate synthetic fertilizer use during beef production propels a vigorous one-way factory-to-ocean nutrient flux, which alternative agriculture models strive to rectify by enhancing in-farm biogeochemical cycling. Livestock, especially cattle, are central to these models, which advocates describe as the context most likely to overcome beef's environmental liabilities. Yet the dietary potential of such models is currently poorly known. Here, I thus ask whether nitrogen-sparing agriculture (NSA) can offer a viable alternative to the current US food system. Focusing on the most common eutrophication-causing element, N, I devise a specific model of mixed-use NSA comprising numerous small farms producing human plant-based food and forage, the latter feeding a core intensive beef operation that forgoes synthetic fertilizer and relies only on locally produced manure and N fixers. Assuming the model is deployed throughout the high-quality, precipitation-rich US cropland (delimiting approximately 100 million ha, less than half of today's agricultural land use) and neglecting potential macroeconomic obstacles to wide deployment, I find that NSA could produce a diverse, high-quality nationwide diet distinctly better than today's mean US diet. The model also permits 70%–80% of today's beef consumption, raises today's protein delivery by 5%–40%, and averts approximately 60% of today's fertilizer use and approximately 10% of today's total greenhouse gas emissions. As defined here, NSA is thus potentially a viable, scalable environmentally superior alternative to the current US food system, but only when combined with the commitment to substantially enhance our reliance on plant food.

## Introduction

Beef is the most resource intensive of all commonly used foods [1–6]. Compared with poultry [5], beef requires 7, 70, 22, and 10 times as much high-quality cropland, total land, irrigation water, and fertilizer, respectively, while emitting 11 times the greenhouse gases per gram of protein. Correspondingly, beef production dominates total US resource use for food production [3,5]. Consuming less beef can thus potentially mitigate climate change and environmental degradation [7–14].

While widely held, this view is not universal [15–25]. When contested, it is most often on the grounds that while the above findings fairly characterize hyperintensive industrial

**Competing interests:** The author has declared that no competing interests exist.

**Abbreviations:** MAD, mean American diet; NSA, nitrogen-sparing agriculture.

agriculture, more sustainable agricultural models that currently contribute very little to total production can deliver far less resource-intensive beef [1,14,26].

Most proposed routes to achieving such putative less resource-intensive beef envision small to medium mixed-use farms, in which cattle are a cornerstone element on account of promoting local recycling of nutrients and thus "soil building" [27,28]. Because on timescales up to a decade the mineral component of soil is approximately fixed, this rather vague idea addresses organic components of soil and thus biogeochemical cycling [29–33]. Indeed, alternative agricultural models are most often promoted on the grounds that they can combat eutrophication and anthropogenic climate change by reducing nutrient leaching and enhancing soil carbon sequestration, respectively [34]. While these goals are often perceived as independent, and soil amendments typically address one or the other [35] (e.g., synthetic fertilizer or biochar for promoting yield or C sequestration, respectively), eutrophication and sequestration are closely interlinked. First, C sequestration rates partly depend on productivity and aboveground litter flux, which rise with rising nutrient availability. Second, while broader than the oceanic Redfield ratio, soil C:N:P:K ratios still reflect the characteristics of their organic inputs, which vary minimally under industrial agriculture. This depresses and narrows agricultural C:N ratios relative to natural ones [36–41], suggesting that biogeochemically preferable proposed alternatives to industrial agriculture stand to both enhance C sequestration by cropland and reduce nutrient leakage and eutrophication.

Soil organic carbon and nitrogen are thus interlinked, and nitrogen—which strongly impacts productivity—also partly controls sequestration in natural and managed environments [28,42]. That is one of the reasons synthetic nitrogenous fertilizer is the most ubiquitous agricultural soil amendment and the prime driver of agricultural eutrophication, the elimination of which is a foundational principle of the suite of promoted practices collectively described as nitrogen-sparing agriculture (NSA) [43,44].

Here, I do not endorse specific agricultural models promoted for potentially minimizing nitrogen imbalances or enhancing carbon sequestration. Rather, I consider a more basic question: Assuming NSA is economically viable (which is beyond the current scope but must be independently shown), can it produce enough nutritious food for 0.3–0.4 billion Americans while lowering food-related resource use? Below, I begin answering this question by quantifying the food output and environmental burdens of a putative rigorously nitrogen-sparing alternative to industrial agriculture, and try to imagine the role it may play in a reconfigured US food and agricultural system [45].

The basic building block of the alternative system is a small mixed-use [42,46] NSA farm whose nitrogen budget is semi-closed. Each such "atom" farm comprises 3 parts (Fig 1). At its core is a small to medium intensive cattle operation (here I only consider beef, but dairy is also possible). Around this cattle core, the farm's available high-quality rainfed cropland comprises 2 subunits. One produces vegetal human food (I use "vegetal" as a shorthand for plant-based food items for human consumption, dominated by grains, fruits, vegetables, and nuts). The other subunit produces served cattle forage (hay, silage, greenchop, and haylage).

The areal extents ("acreage") of each unit farm are unchanged, 1 and approximately 0.43 ha of vegetal and forage production, respectively (Eq S16 and section B in S1 Text), conserving today's approximately 30% of cropland used for feed production. But partitioning of these approximately 1.43 ha to various purposes within each farm can vary in time so as to optimize nutrient dynamics. This may mean, e.g., allowing temporary post-harvest cattle grazing on either of the 2 crop-based subunits, exploiting such marginal resources as minor hay regrowth or aboveground residues, while replenishing nutrient stocks via direct manure deposition. It can also mean deliberate alternating of a given plot of land between vegetal and fodder crops

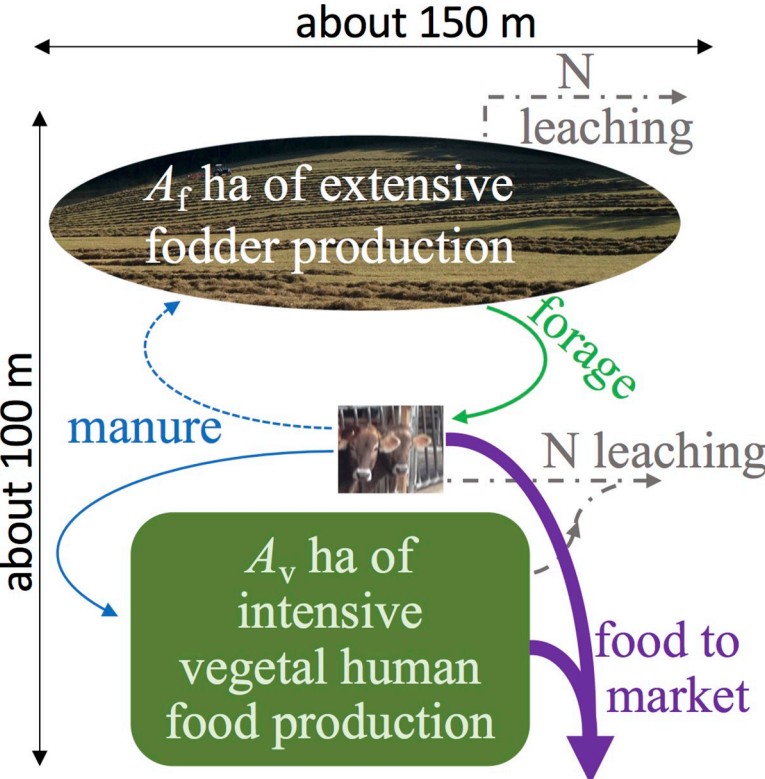

**Fig 1. Schematic representation of the envisioned idealized nitrogen-sparing agriculture arrangement.** It comprises vegetal and forage operational subunits on $A_v$ and $A_f$ ha (1 and approximately 0.43 ha, respectively, shown not to scale; see text for details, with "vegetal" standing for plant-based food items for human consumption), and a core cattle operation occupying negligible area (exaggerated graphically here for presentation clarity). Manure and forage fluxes connect the subunits as shown. The manure flux to the fodder subunit is dashed to highlight that it may or may not be utilized. Nitrogen leaching into the environment is given as dash-dotted gray arrows. Marketed human food is given as downward-pointing purple block arrows.

so as to optimize the role of nitrogen fixers. (But this is made challenging by distinct required bed preparations, compounded by no- or low-till dictates [47].)

Key to successfully deploying the envisioned NSA farms is efficient collection of manure, the only cropland nitrogen source beyond local deposition and fixation. Ensuring this efficiency is the reason the cattle operation I consider here is intensive, relying on served forage rather than grazing, yet this requires on-farm silage siloes or pits and hay storage barns, mowers, rakes, tedders, balers, and front loaders for hay, and mixers/feeders for feeding. Implementing the envisioned NSA farm thus depends on nontrivial capital investment, which highlights the crucial role of policy in a successful widespread transition to NSA. This is not addressed here.

The intensive core cattle operation, whose areal extent is negligible relative to those of the vegetal and forage subunits, interconnects the farm components biogeochemically. I devise a novel mathematical model of the basic farm unit (introduced in detail in sections A and B in S1 Text), and use it (with unique sets of considered crops as described in section B in S1 Text) to calculate the food production output and needed environmental resource input of 250 different randomized (Monte Carlo) implementations of this basic mixed farm, assuming 3 forage qualities. This formalism accounts for geographical and seasonal variability, as well as local preferences and policies. The different assumed forage qualities account for variable farming

know-how, especially as pertaining to optimal crop rotations, as well as variable geographical suitability for forage production. From the simulated food outputs and required resource inputs of these single farm unit implementations, I obtain per hectare input and output statistics of the unit farm for the 3 assumed forage qualities that are expected to characterize on average widespread deployment of the NSA model.

Going beyond the individual farm requires spatial upscaling of these results [48]. The geographical element of this—choosing the candidate area in which NSA farms can be reasonably expected to take hold over time—is straightforward: Because the envisioned agriculture is quite labor- and skill-intensive, it is unlikely to succeed with low yields. I thus focus exclusively on high-quality, precipitation-rich US cropland by summing US Department of Agriculture (USDA) "cropland used for crops" [49] in all contiguous US ("lower 48") states excluding all Southern Plains, Mountains, and Pacific Seaboard states. This is a conservative estimate of suitable area, most notably because it excludes California and the lush Pacific Northwest west of the Cascades. Favoring nonetheless quantitative conservatism over a possibly inflated estimate of NSA's potential, this choice delimits approximately 100 million ha, less than half of today's cropland. This is the area I assume can house the modeled NSA farm.

The economic and broader societal dimensions of upscaling the single NSA farm results are far more challenging [50,51]. If embarked on cavalierly or abruptly, the transition from the current state to the full envisioned NSA system may well lead to supply, demand, and price shocks, as well as dietary and culinary ones, that most will deem intolerable. Avoiding those shocks and rendering the transition successful thus poses important social science challenges. While these challenges are outside of the current geophysical scope, it is my hope that social scientists will address them in future contributions.

With these caveats, I next derive the national-level nutritional and environmental outcomes of a reconfigured US food and agricultural system [45] with the NSA model system deployed on all high-quality US cropland with adequate precipitation for rainfed production, the previously introduced approximately $10^8$ ha. Dividing this high-quality cropland area by the approximately 1.43 ha of the individual farm yields the number of farms it can support. Since the input and output of individual farms is solved for, the total number of farms permitted on the available cropland straightforwardly yields an estimate of the total national food produced and resources required by the new national production model. It is an upper bound estimate, because this scenario is idealized: In reality, deploying the unit NSA farm nationally requires more than simply replicating it everywhere, as discussed briefly earlier, and is likely to be less efficient, delivering less food while requiring more resources. On the other hand, if used wisely, such societal tools as thoughtful legislation and supportive economic and agrarian policies may yield economies of scale that will enhance efficiency and limit the difference between the upper bound output I derive here and the actually realized output.

These limitations notwithstanding, the total food output yields the resultant overall "NSA diet," the beef plus plant output of the NSA model deployed over the 105 million ha of lush US cropland divided by 330 million Americans, i.e., producing no exports, which deviates from the US's status today as a key food and feed exporter. While this is a limitation of the results, the elevated food output of the NSA-based system (reported in the following section) means that it is possible to implement this model, feed the entire domestic population, and have an exported remainder.

The current effort complements an earlier estimate [14] of the amount of "sustainable" beef the US can produce using rangeland alone (but no intensive feed cultivation), as both quantify expected consumption levels under more environmentally desirable production models. But the 2 efforts differ markedly in 2 key ways. First, while earlier we focused on low-productivity western rangelands whose food production potential is limited to beef, here I focus on

high-quality US croplands in eastern and midwestern US states that can be used for any food crop, or deliver over 20-fold more fodder per hectare than arid rangelands [49]. Second, while earlier we considered western rangeland used for grazing, with modest changes relative to today, here I envision a completely novel model—distributed, nitrogen-sparing, co-dependent intensive production of beef and vegetal food in the eastern US.

## Results and discussion

Fig 2 presents the mean annual output of human food from the modeled farm. To put these output measures in perspective, let's compare them to the corresponding US means. The 2017 Census of Agriculture (Table 9 in [52]) estimates that the US has about 364 million ha of total cropland of any kind, of which 260 million is harvested. Over recent years, this enterprise has been producing [53] about 3,800–4,200 kcal and 110–125 g of protein per person per day. Ignoring export–import imbalance, whose impact on the following calculation is smaller than its claimed precision, even considering only harvested land, these values correspond to mean deliveries of about 1.8–1.9 Gcal and 51–58 kg of protein per hectare per year, or—for easy comparison with Fig 2—2.5–2.8 Gcal and 73–83 kg of protein per 1.43 hectare per year. These numbers represent 70%–75% of the caloric and 50%–60% of the protein outputs Fig 2 reports. The considered NSA is thus more, not less, productive than current US agriculture as a whole, even when ignoring altogether the use of non-harvested land (primarily fallow cropland and pastureland, definitely not unimportant resources).

Scaling the single farm results (Fig 2) nationally (with the aforementioned caveats), Fig 3 presents the key nutritional (Fig 3A, 3B and 3E) and environmental (Fig 3C and 3D) outcomes of a national deployment of the NSA. Fig 3A and 3B show the daily per capita beef and protein delivery by the NSA diet. For comparison, the 2019–2020 daily per capita beef intake in the

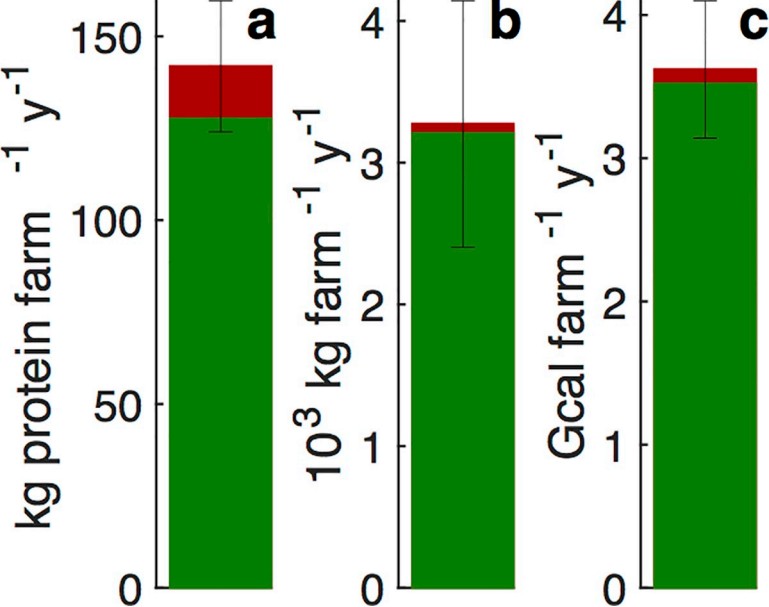

**Fig 2. Annual protein, total food mass, and dietary energy delivered by one approximately 1.43-ha unit farm.** The plant and beef contributions to annual protein (a), total food mass (b), and dietary energy (c) are shown in green (bottom) and red (top), respectively. The whiskers, calculated over the 3 forage qualities and 250 Monte Carlo realizations, span ±1 standard deviation of the total, the plants plus beef sum (the full bar heights). The dietary energy unit in (c) is gigacalories (1 Gcal = $10^6$ kcal). Numerical values are given in S1 Data.

 

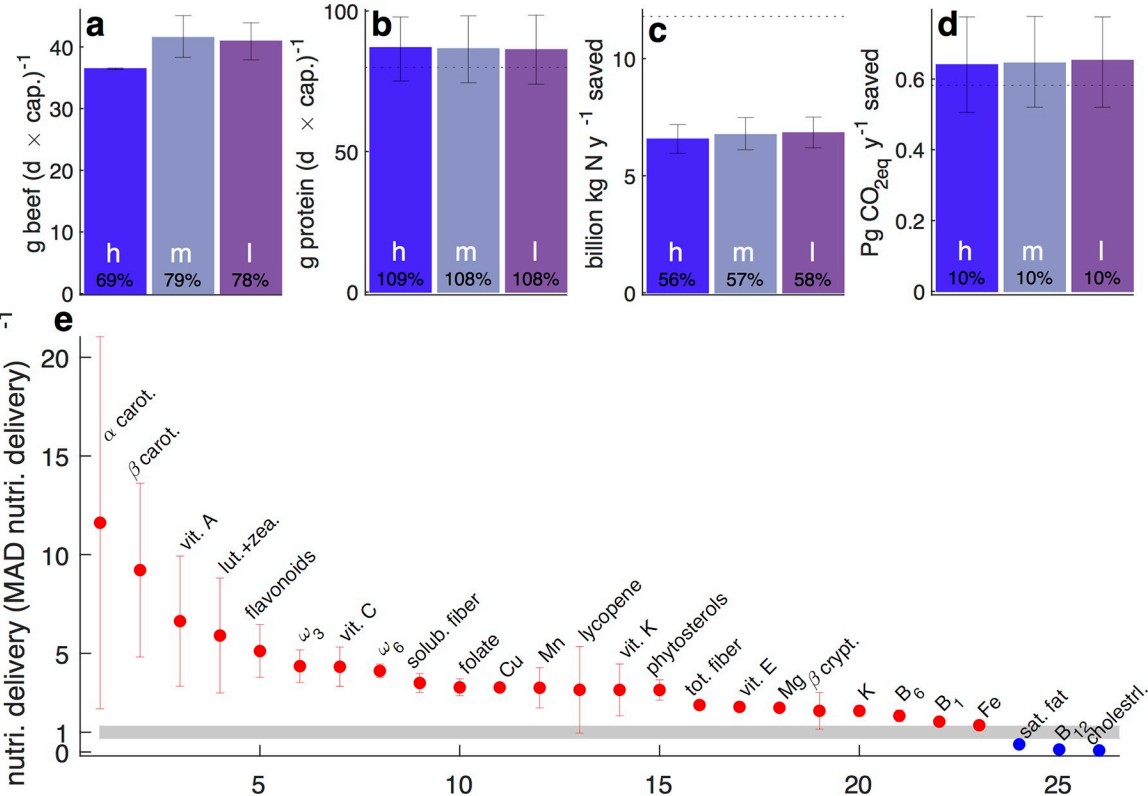

**Fig 3. Nutritional and environmental consequences of repurposing most US high-quality cropland with ≥800 mm·y⁻¹ precipitation (approximately 100 million ha) to the combined vegetal and beef NSA model.** (a–d) Left, center, and right bars denote assuming high (h), medium (m), and low (l) forage quality, respectively (see Tables A and B and sections D–F in S1 Text). (a and b) Annual per capita supply of beef (a) and protein (b). In (b), the dotted horizontal line represents today's (i.e., MAD [1]) national mean per capita protein availability [53]. (c and d) Averted national nitrogen fertilizer use (c) and greenhouse gas emissions (d), the fertilizer mass and emissions that would have been required for producing the same per capita diet using today's conventional agriculture. In (c and d), dotted horizontal lines show today's total national N fertilizer use [54] and agricultural greenhouse gas emissions [55]. The shown numerical percentages inside the bars in (a–d) are relative to national values, in (d) relative to all national greenhouse gas emissions, not only agricultural emissions (the dotted horizontal line in [d]). About 75%–90% of the amounts in (d) are actually saved, because field operations, which are still required in the NSA model, typically account for 10%–25% of emissions [56]. (e) Per capita national mean delivery of specific nutrients by the resultant diet delivered by NSA. Spread whiskers span ±1 standard deviation derived from the 750-member population (3 forage qualities times 250 Monte Carlo realizations for each). Nutrient deliveries are normalized by today's availability in the MAD, e.g., the NSA diet delivers about 11 and 9 times the α and β carotene the MAD offers (the 2 leftmost points). Numerical values are given in S1 Data. cap., capita; carot., carotene; cholestrl., cholesterol; crypt., cryptoxanthin; lut., lutein; MAD, mean American diet; NSA, nitrogen-sparing agriculture; nutri., nutrient; sat., saturated; solub., soluble; tot., total; vit., vitamin; zea., zeaxanthin.

mean American diet (MAD) [1] as recorded by the USDA [53] is 52 g·person⁻¹·d⁻¹. Fig 3E shows mean individual delivery of key nutrients. To permit easy comparison to the current US diet, I present the above nutrient deliveries as fractions of today's MAD [1,53]. For example, the NSA diet delivers on average roughly 11 and 9 times the α and β carotene today's US diet supplies (leftmost points in Fig 3E).

On average, the total (vegetal plus beef) NSA diet delivers 70%–80% and 110% of today's beef and protein consumption (Fig 3A and 3B, respectively), and—because the NSA model relies exclusively on manure—averts 55%–60% of today's nitrogen fertilizer use (Fig 3C).

The NSA model also averts approximately 10% and 110% of today's total and agricultural national greenhouse gas emissions (the former not shown and the latter shown by the dotted line in Fig 3D). The reason the savings exceed 100% is that they reflect elimination of the need

for synthetic fertilizer, the single largest $CO_2$ source in food production (Tables 5–1 and 5–2 of [55]), combined with a national dietary shift toward higher reliance on plants than today's diet. This effect is well above the global shift toward healthier and less environmentally demanding diets predicted by Springmann et al. [57], which makes sense because the dietary shifts they consider are far more modest than the combined agropractical and dietary shifts considered here.

Fig 3E shows some of the more pronounced nutritional impacts of a putative national transition from today's diet to the NSA-based alternative diet. Relative to today's conventional-agriculture-based diet, the NSA diet delivers markedly more of nearly all considered nutrients. Most of the nutrients—phyto-micronutrients, various vitamins, total and soluble fiber, and $\omega_{3,6}$ fatty acids—are protective [7,58,59]. For 3 nutrients, the NSA diet delivers less than today's conventional agriculture diet, but 2 of these—saturated fat and cholesterol—are considered nutritionally deleterious [60,61], although this view is being actively refined [62]. The only clearly negative nutritional impact of the NSA diet is the reduced vitamin $B_{12}$ intake, which primarily reflects the replacement in the NSA diet of poultry, pork, dairy, and eggs with plant alternatives, reducing the overall prominence of animal-based food in this diet. While this is potentially alarming in light of the critical neurological importance of $B_{12}$, the deficiency is easy to overcome [63].

The answer to the basic question motivating this inquiry—can nutrient-autonomous NSA domestically produce enough nutritious food for all Americans—is thus an emphatic yes. Even with stringently limiting nitrogen sources to manure, atmospheric deposition, and symbiotic fixation, limiting agriculture to precipitation-rich regions, and allowing for a 15% yield penalty in the NSA model relative to today's conventional model (see sections B and N in S1 Text for justification of and sensitivity to this yield penalty [64]), NSA produces more than enough food (including protein) for all Americans, at considerable environmental savings, while markedly improving the mean national diet and requiring a fairly modest reduction in beef consumption (Fig 2A).

Because the above results hold with ample margins, they are likely to persist even if some of my assumptions are revisited and relaxed.

These results are possible mostly because the model only produces beef and vegetal food, but no poultry, pork, dairy, or eggs. In a more realistic rendition of the model whose implementation chances are higher, this limitation will have to be eliminated or relaxed. The key reason the model can so bountifully feed the populace is the spontaneous emergence in the model of a plant-heavy diet despite allocating 6-fold more land to beef than to any single plant item (0.43 versus 0.07 ha). Because plants in general require far less land, much more plant protein can be produced under the NSA model, even on the fairly small portion of total US cropland I exploit here, and even with allocating nearly a third of all that cropland for beef production.

Because I model NSA with beef at its core, and because the output proves more than sufficient, permitting most of today's beef consumption, some readers may view the results as a waiver of beef's environmental liabilities. This is negated by beef's modest positive and disproportionately negative contributions to the NSA diet, and by its outsized land use, but is supported by its enhancement of vegetal yields.

To evaluate the net effect of this duality, let's compare protein delivery by the main model (Fig 3B) to delivery of no beef, forgoing manure benefits but having potentially more land for vegetal production (Fig 4). The left (dark green) bar corresponds to rewilding in each farm the 0.43-ha forage area—30% of all land—and using the original 1 ha of vegetal production. On the right (light green), the production model now occupies the full area, 1.43 ha per farm. In both alternative scenarios, vegetal productivity is significantly reduced because of the absence

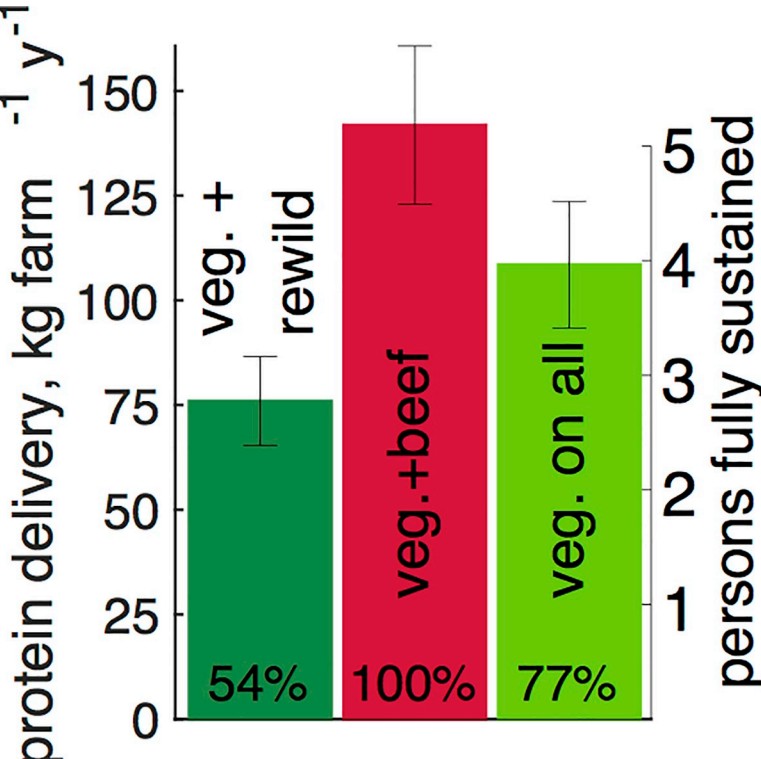

**Fig 4. Daily per capita protein availability in 3 scenarios.** The middle bar (red; "veg. + beef") is the average of the 3 bars in Fig 2B. This average protein delivery—the red bar and this paper's main result—is the reference here (hence "100%"). The left bar (dark green; "veg. + rewild") envisions in each farm rewilding the 0.43-ha forage subunit and using the 1-ha vegetal subunit for producing plant items for human consumption. This option delivers 54% of the full per capita protein availability of the reference option (in which the 0.43 ha is used for forage and thus beef production). The right (light green; "veg. on all") bar envisions using the full 1.43 ha for vegetal human food production, yielding 77% of the reference protein delivery. The right vertical axis shows the number of individuals (assumed to consume 75 g of protein per person per day) whose protein needs can be fully met by the considered farm, ranging from about 2.8 to 5.1 adults. Numerical values are given in S1 Data.

of manure. Comparing the central and left bars shows that eliminating cattle and rewilding 30% of the considered land delivers approximately 50% of the available protein. Comparing the central and right bars shows that preserving the main model's total land area while forgoing manure benefits also lowers protein delivery, to approximately 75% of the reference protein availability. By significantly enhancing vegetal output, beef is thus a key component of the NSA model, despite directly yielding only a modest fraction of total protein (Fig 2A).

At least in principle, NSA offers a realistic alternative to today's industrial model, provided the US population is ready to replace much of the animal-based portion of its diet with plant-based alternatives. Beef cattle have a uniquely important contribution to making this transition possible and palatable.

## Materials and methods

I consider a specific vegetal food and beef coproduction NSA model, and quantify its potential productivity and role in a thoroughly reconfigured US food system. Clearly, more efficient livestock [5,17,65] could also enhance nitrogen availability for vegetal production beyond symbiotic fixation and atmospheric deposition, while exacting lower environmental impacts than beef [65]. I nonetheless focus on beef for 2 reasons. First and foremost, the elemental

composition of beef manure is nearest (among the major livestock types) to those of crops [38], making it agronomically superior. Second, beef is deeply entrenched and hard to replace [66] in worldwide diets.

The basic structure of the modeled farm is guided by a key tenet of NSA, maximizing local nutrient recycling and minimizing local nutrient imbalances. Heeding this adage while focusing on nitrogen as a proxy for limiting nutrients in general, the modeled farm receives only atmospheric sources of the key environmental variables limiting land productivity—precipitation and deposition plus symbiotic fixation for water and reactive nitrogen, respectively. Obviously, complete models of NSA will have to also address supplies of phosphorus, potassium, and other agriculturally vital nutrients whose cycles differ markedly from that of nitrogen, as well as latitude and length of growing season, seasonal patterns of precipitation, cloud cover, temperature, and humidity among other limiting factors. The current calculation is thus an upper bound estimate in the sense that in some locales, limitations posed by other variables can suppress local productivity below the levels envisioned here.

The modeled farm comprises 3 elements shown schematically in Fig 1. Production of vegetal food for human consumption and forage for cattle claim $A_v$ and $A_f$ ha, respectively. Reactive nitrogen sources include deposition and symbiotic fixation, augmented by manure from the cattle operation. The latter contribution is determined by the herd size and composition (sections E and F and Table B in S1 Text) and by diet composition (section D and Table A in S1 Text).

To maximize manure recovery and yields, the forage subunit is mostly not directly grazed, delivering served cattle fodder (greenchop in summer and hay, haylage, or silage otherwise) instead. The core cattle operation is thus nutritionally grass-based yet operationally mostly an intensive, spatially confined operation whose area is negligible relative to $A_v$ or $A_f$.

The model's governing equations are simple nitrogen conservation statements for the 3 subunits,

$$y_v = \rho_v(d + f_v + \gamma\alpha n_m A_V^{-1}) \tag{1}$$

$$y_f = \rho_f[d + f_f + (1 - \gamma)\alpha n_m A_f^{-1}] \tag{2}$$

$$n_m = \beta A_f y_f, \tag{3}$$

addressing the vegetal, forage, and cattle subunits, respectively. Above $y_v$ and $y_f$ are the nitrogen yields of the vegetal and forage subunits (kg N·ha$^{-1}$·y$^{-1}$), $n_m$ is the number of mother cows the herd comprises, $\rho_v$ and $\rho_f$ are the corresponding dimensionless nitrogen retention fractions (the productive fraction of all available nitrogen, parameterized as in section B1 in S1 Text), $d$ and $f_v$ and $f_f$ are rates of nitrogen introduction into the respective subunits by atmospheric deposition and symbiotic fixation, and $\gamma$ is the manure partitioning parameter, with $\gamma = \{0,1\}$, respectively, corresponding to all the manure delivered to the vegetal and forage subunits. Nitrogen cycling efficiencies are given by $\alpha$ and $\beta$. Annual production of plant available manure nitrogen by 1 cow and her associated animals (e.g., roughly 0.7 calf·y$^{-1}$ and 0.04 bull; see section E in S1 Text for the herd structure) is given by $\alpha$, while cattle need for forage nitrogen in cow-years per kilogram forage N is given by $\beta$.

Even though the equations are flexible enough to accommodate any $\gamma = [0,1]$, section A1 in S1 Text shows that $\gamma = 1$ is optimal and is thus the default value used. For most applications,

therefore

$$y_v = \rho_v(d + f_v + \alpha n_m A_V^{-1}) \tag{4}$$

$$y_f = \rho_f(d + f_f) \tag{5}$$

$$n_m = \beta A_f y_f, \tag{6}$$

which combine into

$$y_v = (d + f_v) + \alpha \beta \rho_v \rho_f A_f A_v^{-1}(d + f_f). \tag{7}$$

The land areas $A_v$ and $A_f$ and their ratio are key free parameters of the system not constrained by the governing equations. They can therefore be chosen so as to, e.g., maximize protein or vitamin $B_{12}$ delivery, or minimize greenhouse gas emissions. Because the current contribution is a preliminary conceptual inquiry, not an effort to derive the definitive model for optimal NSA, I choose $A_f/A_v = 3/7 \approx 0.43$ (section B in S1 Text), conserving the value of this ratio in today's US food system. Given the notorious difficulties of dietary modifications of most any kind [67], compounded by limited willingness for personal sacrifice in the name of environmental betterment, minimizing deviation from today's state seems reasonable. Follow-up work can revisit these choices so as to optimize various objectives. The basic atom farm modeled here thus occupies approximately 1.43 ha.

The herd comprises mature mother cows and bulls, the necessary replacement animals for both categories, and output calves and steers (section E and Table B in S1 Text). Numbers of all animal categories are fixed proportions of the number of mother cows $n_m$, e.g., 4 bulls or 16 replacement heifers per 100 cows. Consequently, the herd structure and size, and therefore forage requirements and beef and manure output, linearly track $n_m$, whose quantification (section B in S1 Text) is a key element of the solution. I solve the governing equation, Eq 7, in a Monte Carlo formalism, with $A_f = 1$ ha, $A_v \approx 0.43$ ha, $\rho_v$ and $\rho_f$ parameterized as functions of available nitrogen inputs as discussed in section B1 in S1 Text, and all remaining right-hand-side terms randomly drawn from empirically derived distributions. With $y_v$ thus evaluated and $\rho_f$ known (by parametrization valuation), I use Eqs 5 and 6 to get $y_f$ and $n_m$, respectively. All remaining characteristics of the solution follow from $\{y_v, y_f, n_m\}$ thus evaluated. The single farm results can then be applied to larger areal, geographical extent, subject to the caveats and precautions highlighted above and in section C in S1 Text.

In this preliminary stage of this work, once an applicable geographical extent of $A_{up}$ ha is selected for the upscaling, that area is simply assumed to be fully dedicated to NSA, i.e., to house $N_{farm} = A_{up}/1.43$ farms. With the single farm solution outputs and inputs, and this $N_{farm}$, the environmental and nutritional consequences (section B and Eqs S18 and S19 in S1 Text) of the transition to the NSA model for 330 million Americans consuming the vegetal and beef output of this limited-area NSA deployment are straightforwardly obtained.

The system's outputs—vegetal food, beef, and manure—all depend on the envisioned cattle diet. The considered herd subsists mostly on variable-quality [68] forages, reflecting the broad range of soils and climates in US croplands. In an effort to enhance output while most closely tracking today's practices [3,5], the herd diet also comprises 5% by mass (roughly today's fraction) of energy- and protein-rich agroindustrial byproducts (e.g., millfeed from grain milling, sugar beet pulp, or citrus peel from juice production) that are widely used as cattle feeds today. For any specified cattle diet (ranging in the forages and byproducts used; Table A in S1 Text), I use well-established equations [68,69] to predict the herd's forage consumption, weight gain, and manure nitrogen excretion (sections D–G in S1 Text).

## Supporting information

**S1 Data. A text data file reporting all the numerical values of main text and supplementary figures.**
(DAT)

**S1 Text.** Supplementary information, including supplementary sections A–N, Eqs S1–S32, Tables A and B, and Figs A–F.
(PDF)

## Acknowledgments

I heartily thank the Lamont Doherty Earth Observatory at Columbia, and the Department of Earth and Planetary Sciences at Harvard, for their warm hospitality during my extended adjunct affiliations. By providing library and computational support, both materially aided this effort. I also thank William H. Schlesinger of the Cary Institute for his careful reading of an early draft and his thoughtful comments on it, and the EUS program at Bard for their generous support.

## Author Contributions

**Conceptualization:** Gidon Eshel.

**Data curation:** Gidon Eshel.

**Formal analysis:** Gidon Eshel.

**Funding acquisition:** Gidon Eshel.

**Investigation:** Gidon Eshel.

**Methodology:** Gidon Eshel.

**Project administration:** Gidon Eshel.

**Resources:** Gidon Eshel.

**Software:** Gidon Eshel.

**Supervision:** Gidon Eshel.

**Validation:** Gidon Eshel.

**Visualization:** Gidon Eshel.

**Writing – original draft:** Gidon Eshel.

**Writing – review & editing:** Gidon Eshel.

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
