## [Editor Report · Decision Letter 0]

18 Oct 2020

Dear Gidon, 

Thank you for submitting your manuscript entitled "Toward Plugging Nutrient Leakage in U.S. Agriculture" for consideration as a Research Article by PLOS Biology.

Your manuscript has now been evaluated by the PLOS Biology editorial staff, as well as by an academic editor with relevant expertise, and I'm writing to let you know that we would like to send your submission out for external peer review. Sorry for the slow response while we waited for external advice.

Please re-submit your manuscript within two working days, i.e. by Oct 20 2020 11:59PM.

Kind regards,

Roli

Senior Editor

PLOS Biology

---

## [Decision Letter · Decision Letter 1]

14 Dec 2020

Dear Gidon,

Thank you very much for submitting your manuscript "Toward Plugging Nutrient Leakage in U.S. Agriculture" for consideration as a Research Article at PLOS Biology. Your manuscript has been evaluated by the PLOS Biology editors, an Academic Editor with relevant expertise, and by three independent reviewers; I'm afraid we weren't able to engage the reviewer who had seen this at another journal.

IMPORTANT: You'll see that the reviewers are broadly positive about your study, but each raises a number of concerns that need to be addressed. In addition, the Academic Editor has kindly provided some guidance as to how you should address the reviewers' comments; s/he wrote it with the idea of me paraphrasing it in the letter, but actually I think it's better if I transmit it verbatim - see the foot of this email. In particular we draw your attention to the requests for further analyses (including sensitivity analyses), greater clarity regarding the modelling, and overall accessibility, especially with our broader readership in mind.

In light of the reviews (below), we will not be able to accept the current version of the manuscript, but we would welcome re-submission of a much-revised version that takes into account the reviewers' comments. We cannot make any decision about publication until we have seen the revised manuscript and your response to the reviewers' comments. Your revised manuscript is also likely to be sent for further evaluation by the reviewers.

We expect to receive your revised manuscript within 3 months. 

**IMPORTANT - SUBMITTING YOUR REVISION**

*Re-submission Checklist*

*Published Peer Review*

*PLOS Data Policy*

*Blot and Gel Data Policy*

Best wishes,

Roli

Senior Editor,

rroberts@plos.org,

PLOS Biology

REVIEWERS' COMMENTS:

Reviewer #1:

[identifies himself as Gilles Billen]

 I liked very much this paper, as well as the previous ones by the same author: their provocative prospective views of what alternative food systems could look like are very interesting and stimulating. The concept of « nutrient sparing agriculture » (NSA) with the objective of zero industrial fertilizer use and full closure of nutrient cycles is quite useful. 

The demonstration that the generalization of this model offers a realistic alternative to today's industrial agriculture is quite convincing.

I am impressed by the degree of details in the link the author is able to establish between his hypothetic agricultural system and the diet, described at the level of micronutrients. 

I have however some objections to the formulation of the model, from an agronomical point of view.

(1) The author seems to give no consideration to crop rotations. In the "v" part of the farm, the different crops are distributed randomly, and independently of each other. Their Nyields are defined a priori, as constant values, independently on the other, and on the applied fertilization. Similarly, in the separated "f" part of the farm, symbiotic fixation is constant for a given forage quality. In the real world, fodder crops and the different "v" crops alternate on the same land, and that is how nitrogen fixing plant can sustain the growth of the succeeding crop (in addition to manure transfers). Here, if I understood correctly, yv is determined by the basket of vegetal production and this determines the amount of manure required.

(2) N inputs through symbiotic fixation are all but natural inputs: these are the levers for intensification of the system, and not a natural constraint as considered in this model.

(3) The model assumes linearity of total Nyield (vf and yv) versus total soil Ninputs (equ S1 and S2), and discusses the proportionality (or leakage) coefficient (0.8 - 09) as constant N use efficiency . Yet in the real world, the yield fertilization relationship is not linear, and tends to plateau at higher rate of N inputs. This is the cause of increased environmental losses for intensive systems: the leakiness of a cropping system is a function of its level of N inputs. The assumption of linearity of the equation prevents a correct estimate of the environmental losses which are at the core of the paper. 

(4) This is more a sentimental objection: I hate the idea of confined livestock farming! Outdoors grazing is, and I hope will remain, an essential part of our traditional landscapes…. 

Reviewer #2:

This paper presents the results of a modeling exercise that calculates the optimal farm size for minimizing nitrogen use (N sparing agriculture) then scaled to enough arable cropland in the US to meat American dietary needs. The model farm is highly simplified to 3 components: forage production, food production, and animal production. Nitrogen is spared by N fixation sufficient to replace N exported in crop and beef harvests plus within-farm recycling of that used for animals via manure. Crops are random mixtures of those grown in the US chosen to meet human nutritional requirements. Environmental N losses are based on nitrogen use efficiencies of 80% for food crops and 90% for forage. The results of monte carlo simulations support many other analyses showing that reactive N losses can be minimized by moving populations to a plant-based diet. What is unique is its also doing so by reducing beef production by only 20-40% rather than 100%. However the highly simplified nature of the analysis begs the significance of this finding and there needs to be a more robust analysis of the factors that allow this inclusion.

This is, in fact, is a hard paper to read: it's not well structured and there is a lot of opinion and loose rationalizing and general wordiness in the narrative, overuse of references, insufficient information on nitrogen fluxes, and missing figure legends. In other words, it needs a good editing. 

That said, to me the chief scientific weakness appears related to N use efficiencies, central to the paper's overall conclusions and unrealistically high for even organic systems like these. Choosing average values between 50% (typical for fertilized field crops) and 100% (based on Hubbard Brook, an aggrading forest) isn't sufficient without further quantitative justification. Values should be based on average (not cherry picked) literature values, area-weighted for the different types of cropping systems. Moreover, most of the food crops in these systems appear to be horticultural - with NUEs often well below 50% regardless of N source.

I should add that if not already, the paper should be reviewed by a modeler who can fully evaluate the derviced equations in supplemental material.

Specific comments

1. throughout, the word "vegetal" is an odd usage to denote food crops, i.e. used for direct human consumption. I've been reading the agricultural literature for decades and this is the first time I've encountered it.

2. the Abstract is 50% introduction. Better to spend the space describing the model and its major assumptions.

3. line 71. Figure 1 seems almost unnecessary in its simplicity. The image of CONUS and center pivot irrigation circles are confusing - the modeling is not geospatial and in any case not irrigated; better to simply represent the 3 farm elements with some explanatory arrows denoting major nitrogen flows.

4. line 65 - what does sequestration limiting mean. Nitrogen per se has little direct impact on carbon sequestration if that's what is meant.

5. line 115. What exactly is 1.43 ha - the entire individual farm unit? 

6. line 261. The modeling is central to the paper and deserves describing in Methods, not buried in the supplemental material.

Reviewer #3:

The paper claims that substantial environmental benefits (reduction in nutrient loss, greenhouse gas emissions) could arise from a shift towards NSA (Nitrogen Sparing Agriculture) in the United States. While these claims are not novel (the author cites several papers that make the case), as far as I am aware the national perspective, for the USA, is novel. I am very sympathetic to the general idea - from an economic perspective it is my view that we - I am writing in Europe - are too far to the right on our agricultural response curves, particularly for manufactured nitrogen. I cannot comment on the USA, except to note that I assume that you continue to be 'land rich' and thus intensity per acre will be lower for fertilisers than in Europe. However, the argument probably still holds. The paper would be of interest to academics and policy makers as it takes farm level effects of attempting to close nutrient systems - essentially, specialise in (intensive) beef production and use manure as fertiliser for a range of field and tree-based vegetables and fruits - to the national level. If this less leaky system 'works' at the farm level, why not extend it? This is also the main flaw in the paper - if the system results in a vector of environmental effects on one acre, the effect of many acres is just a multiple of that vector of effects. If only it were that easy! For example, I could ask, from a UK perspective, what is the national environmental effect if all farms recycled manure, grew vegetables, replaced orchards, grew cover crops, reduced nitrogen use per hectare on wheat to circa 100kg N per ha (this is low for the UK) … ? We do know very clearly that producing less ruminant-based livestock (beef and sheep in particular) and eating more plants will have quite large beneficial effects; personally I think the GHG effects are over-emphasised and nutrient-cycling benefits are under-emphasised: agriculture is still a relatively small emitter of GHGs in western economies and the potential negative trade-offs of even small reductions in food supply are high, both in terms of probability of effect and level of effect. However, nutrient loss through leaching of nitrogen and run-off of phosphorous are major land-based issues and, as the author shows, correlated with GHG emissions. Thus, my major criticism of the aggregate NSA scenario is that at best it is not sufficiently interesting and at worst it could lead policy makers in a very dangerous direction. While it is stated that an economic analysis is not part of the work I do think that there should be greater acknowledgement that there is an underlying system that cannot just be swapped for the modelled system without substantial disruption. In essence 'industrial' - to use the author's term - agriculture has delivered cheap food that has freed society to do things other than growing food and, as importantly, freed society from worrying about where food will come from each week. Yes, the environmental cost of this is large and needs addressing; however, I think it is dangerous just to present an end-result without considering the transition to that result. Indeed, a novel paper would address how we move to a better place whilst also addressing the effects of this transition on e.g. consumer food prices.

Regarding the analysis itself, I am uncomfortable with the format for the given topic. While 'Results' followed by 'Methods' is standard for Plos Biology it does make it hard for the reader. For example, Monte Carlo Analysis is mentioned without any explanation of what it is or why it is used. There is a lack of transparency about how nutrient loss is calculated - equations are mentioned but we have to take it on trust that the results are valid. Furthermore, interventions in agriculture designed to improve environmental outcomes are inherently variable: they work, they don't work, they work but then it rains, they don't work but then it rains and they do work… and so forth. 

This isn't very helpful and as I wrote at the start, I do think there is important work to be done here. I would like to see the workings behind the models, some sensitivity analysis - just generally to understand better how the proposed NSA system works on farm and for the environment (one of the questions I am asked as a reviewer is: "Are details of the methodology sufficient to allow the experiments to be reproduced?" - to which I would have to answer 'no'). Perhaps this important work would be more suited to a different type of journal.

COMMENTS FROM THE ACADEMIC EDITOR [lightly edited]:

I agree with your way forward [Major Revision] but equally share your [my!] concern about something coming back half-baked. Reviewer #2 is quite spot-on for me that it's not the easiest paper to read.

So how to lend some more direction? Technically, there are I think quite serious concerns about the appropriateness of the N use efficiencies raised by Rev #2 that should be explored with further analysis. This is where the sensitivity analyses from Rev #3 could partly come in. Rev #3 also raises a major concern about the scaling of the system being linear (a "vector") and whether that is appropriate. Again, do we need further (sensitivity?) analysis here?

Beyond this modelling details, both Revs #2 and #3 lament the way the workings of the modelling and underlying equations / assumptions are presented. More detail and precision is clearly needed in a revision and that should be a straightforward point to communicate to the author. Finally, I think the points raised by Rev #3 about the underlying economics are important to acknowledge in a meaningful way. Less sure about how this one looks, but maybe a discussion of the limitations of the current approach w.r.t. economics might be sufficient?

---

## [Decision Letter · Decision Letter 2]

20 Apr 2021

Dear Gidon,

Thank you for submitting your revised Research Article entitled "Toward Plugging Nutrient Leakage in U.S. Agriculture" for publication in PLOS Biology. I have now obtained advice from two of the original reviewers and have discussed their comments with the Academic Editor. 

Based on the reviews, we will probably accept this manuscript for publication, provided you satisfactorily address the remaining points raised by the reviewers. Please also make sure to address the following data and other policy-related requests.

IMPORTANT:

a) Please address the remaining concerns raised by reviewer #2.

b) Reviewer #2 makes a suggestion regarding the title of your article. We agree with this, as it makes it immediately more explicitly informative. However, we also wonder whether it would be better to truncate their suggested title thus: "Small-scale integrated farming systems can abate continent-scale nutrient losses." Removing the "in US agriculture" would broaden the appeal of the article, and you already make it abundantly clear in the Abstract that you're specifically considering the U agricultural system, so readers will not be misled. If you're not comfortable doing this, then stick with the reviewer's suggestion.

c) Please attend to my Data Policy requests further down. Specifically, we need you to provide the numerical values for Figs 2ABC, 3ABCDE, 4, S2, S5, S6, and cite the location of the data clearly in the relevant Figure legends.

d) A tiny point, but I note that Fig 3E doesn't currently have a label.

We expect to receive your revised manuscript within two weeks. 

*Published Peer Review History*

*Early Version*

Best wishes,

Roli

Senior Editor,

rroberts@plos.org,

PLOS Biology

DATA POLICY:

Regardless of the method selected, please ensure that you provide the individual numerical values that underlie the summary data displayed in the following figure panels as they are essential for readers to assess your analysis and to reproduce it: Figs 2ABC, 3ABCDE, 4, S2, S5, S6. NOTE: the numerical data provided should include all replicates AND the way in which the plotted mean and errors were derived (it should not present only the mean/average values).

DATA NOT SHOWN?

REVIEWERS' COMMENTS:

Reviewer #1:

[identifies himself as Gilles Billen]

I am fully satisfied by the responses of the author to my comments on the earlier version of the ms. He brought major changes in the paper which improved its value and readibility a lot.

Reviewer #2:

The revision is substantially improved, I appreciate the time an attention the author put into consideration of my comments. The paper is still hard to read in places and deserves a rigorous language editing for clarity, brevity, and organization, but the overall rationale, approach, and interpretation are much more understandable than in the earlier version. That said, additional attention is warranted:

1. the title should better reflect the paper content to allow the reader to anticipate its content. "Towards plugging nutrient leakage" is clever but cryptic, better to save for a general readership or lay outlet. There are, in fact, hundreds of ways to plug nutrient leakage so some specificity is warranted. Consider something like Small-scale integrated farming systems can abate continent-scale nutrient losses in US agriculture.

2. the abstract is still 1/3 introduction (everything up to line 14), not abstract. Reduce to 2-3 lines and use the saved space to build out the "how" question on line 20: "….NSA could produce a diverse….." There is no information about how these results are achieved, and this deserves some mention (e.g. "Converting 110 Mha to 1.4 ha units with xx ha devoted to plant-based food, xx ha devoted to forage, and substituting legumes for synthetic fertilizer etc.…." or the like). 

3. lines 37-52. The logic of this paragraph escapes me - what is the main point, that soil building is a fool's errand? That C:N:P:K ratios matter? That SOC and nitrogen are interlinked (following sentence) - if the latter, this is a truism in soil biology so doesn't really deserve a paragraph of pre-defense. Anyway, the paragraph is a head-scratcher and needs clarification or deletion (little would be lost by deleting I think).

4. line 54. "…also partly controls sequestration…" requires explanation, and I'm not aware of any environments "primarily devoted to carbon sequestration" (next line) - is this a potshot at regenerative agriculture or an oblique reference to old growth forests or….?

5. line 57. in what way is synthetic N fertilizer the foundational principle of NSA - this seems counter to your argument, or maybe it's a sentence structure issue.

6. lines 74-82. Would it be possible to provide a crisp one sentence description of the paper's objective. This objectives paragraph (my inference) is unreasonably wordy.

6. line 83-176. There is a major structural issue with the paper that starts about here. We're still in the (very long) Introduction, but beginning with line 83 we're told how the analysis was performed. I understand that PLOS-Bio requires Methods at the end, and I understand that you are trying to address earlier reviewer comments to address the awkwardness of this arrangement, but a heading is needed to break a very long narrative. If the journal would allow it, maybe start a section called Approach or if not maybe use sub-headings in the introduction (Objectives, Local modeling, National modeling).

7. line 105. typo

8. lines 150-176 are an odd mix of approach and discussion (interpretation of the approach and its limitations with reference to results) - it would greatly improve readability if all of these cautionary notes and assumptions were presented after the results as in standard scientific writing where interpretation of results is placed in the discussion.

9. line 161. The lack of exports is important but hard for a naïve reader to put in context - please note how many acres are currently used to support contemporary exports. 25 Mha?

10. line 174. What does earlier refer to - there is no rangeland grazing considered up to this point, and why does the voice change from singular to plural.

11. line 210. typo (figure 3 not 2).

12. line 212-215. This seems like a non-sequitur to me - why can't poultry, dairy, and pork substitute for beef in the protein production targets, and thereby also save fertilizer inputs - is it because poultry and pigs can't eat forage, and if grain crops were substituted for forage the balance would be lost? The statement deserves better explanation.

13. throughout - this is semantic, but I continue to have a problem with the word vegetal, and though plants for human consumption is more of a mouthful, it's the common way to describe what you (and M-W, surprisingly) call vegetal. If you're striving for readability I suggest substituting.

---

## [Editor Report · Decision Letter 3]

5 May 2021

Dear Gidon,

On behalf of my colleagues and the Academic Editor, Andrew Tanentzap, I'm pleased to say that we can in principle offer to publish your Research Article "Small-scale integrated farming systems can abate continental-scale nutrient leakage" in PLOS Biology, provided you address any remaining formatting and reporting issues. These will be detailed in an email that will follow this letter and that you will usually receive within 2-3 business days, during which time no action is required from you. Please note that we will not be able to formally accept your manuscript and schedule it for publication until you have made the required changes.

IMPORTANT: There are a few changes that I'll need you to make regarding your data provision, but in the interests of expediency I'll pass on these requests to my colleagues who will handle the next stage. However, I'll just state them here so you're aware. Essentially, I see that you've provided the underlying data as a downloadable flat-text file on ResearchGate. This is OK-ish, but you should therefore ideally provide the URL in the Figure legends, i.e. "Numerical values are given in https://www.researchgate.net/publication/351156992." And alternative would be to upload this flat-text file (or an Excel version of it) as "S1_Data" and then change the citation in the Figure legends to "Numerical values are given in S1 Data" (this will then be linked to the file in the final manuscript). Either of these approaches would be good. I suspect that you will also be asked to rename your Supp Fig files to S1_Fig.pdf, S2_Fig.pdf, etc...

PRESS: We frequently collaborate with press offices. If your institution or institutions have a press office, please notify them about your upcoming paper at this point, to enable them to help maximise its impact. If the press office is planning to promote your findings, we would be grateful if they could coordinate with biologypress@plos.org. If you have not yet opted out of the early version process, we ask that you notify us immediately of any press plans so that we may do so on your behalf.

Thank you again for supporting Open Access publishing. We look forward to publishing your paper in PLOS Biology. 

Sincerely,

Roli 

Roland G Roberts, PhD 

Senior Editor 

PLOS Biology